# Incidence of an Intracellular Multiplication Niche among *Acinetobacter baumannii* Clinical Isolates

Tristan Rubio,[a] Stéphanie Gagné,[a] Charline Debruyne,[a] Chloé Dias,[a] Caroline Cluzel,[b] Doriane Mongellaz,[a] Patricia Rousselle,[b] Stephan Göttig,[c] Harald Seifert,[d,e] Paul G. Higgins,[d,e] Suzana P. Salcedo[a]

[a]Laboratory of Molecular Microbiology and Structural Biochemistry, Centre National de la Recherche Scientifique UMR5086, Université de Lyon, Lyon, France
[b]Laboratoire de Biologie Tissulaire et d'Ingénierie Thérapeutique, UMR5305, CNRS, Université de Lyon, France
[c]Institute for Medical Microbiology and Infection Control, University Hospital, Goethe University Frankfurt am Main, Frankfurt, Germany
[d]Institute for Medical Microbiology, Immunology and Hygiene, Faculty of Medicine and University Hospital Cologne, University of Cologne, Cologne, Germany
[e]German Centre for Infection Research (DZIF), Partner Site Bonn-Cologne, Cologne, Germany

Tristan Rubio and Stéphanie Gagné contributed equally. The order of the authors was decided with mutual agreement by the number of figures.

**ABSTRACT** The spread of antibiotic-resistant *Acinetobacter baumannii* poses a significant threat to public health worldwide. This nosocomial bacterial pathogen can be associated with life-threatening infections, particularly in intensive care units. *A. baumannii* is mainly described as an extracellular pathogen with restricted survival within cells. This study shows that a subset of *A. baumannii* clinical isolates extensively multiply within nonphagocytic immortalized and primary cells without the induction of apoptosis and with bacterial clusters visible up to 48 h after infection. This phenotype was observed for the *A. baumannii* C4 strain associated with high mortality in a hospital outbreak and the *A. baumannii* ABC141 strain, which was isolated from the skin but was found to be hyperinvasive. Intracellular multiplication of these *A. baumannii* strains occurred within spacious single membrane-bound vacuoles, labeled with the lysosomal associate membrane protein (LAMP1). However, these compartments excluded lysotracker, an indicator of acidic pH, suggesting that *A. baumannii* can divert its trafficking away from the lysosomal degradative pathway. These compartments were also devoid of autophagy features. A high-content microscopy screen of 43 additional *A. baumannii* clinical isolates highlighted various phenotypes, and (i) the majority of isolates remained extracellular, (ii) a significant proportion was capable of invasion and limited persistence, and (iii) three more isolates efficiently multiplied within LAMP1-positive vacuoles, one of which was also hyperinvasive. These data identify an intracellular niche for specific *A. baumannii* clinical isolates that enables extensive multiplication in an environment protected from host immune responses and out of reach of many antibiotics.

**IMPORTANCE** Multidrug-resistant *Acinetobacter baumannii* isolates are associated with significant morbidity and mortality in hospitals worldwide. Understanding their pathogenicity is critical for improving therapeutic management. Although *A. baumannii* can steadily adhere to surfaces and host cells, most bacteria remain extracellular. Recent studies have shown that a small proportion of bacteria can invade cells but present limited survival. We have found that some *A. baumannii* clinical isolates can establish a specialized intracellular niche that sustains extensive intracellular multiplication for a prolonged time without induction of cell death. We propose that this intracellular compartment allows *A. baumannii* to escape the cell's normal degradative pathway, protecting bacteria from host immune responses and potentially hindering antibiotic accessibility. This may contribute to *A. baumannii* persistence, relapsing infections, and enhanced mortality in susceptible patients. A high-content micros-copy-based screen confirmed that this pathogenicity trait is present in other clinical

Address correspondence to Suzana P. Salcedo, suzana.salcedo@ibcp.fr.

The authors declare no conflict of interest.

*A. baumannii* isolates. There is an urgent need for new antibiotics or alternative antimicrobial approaches, particularly to combat carbapenem-resistant *A. baumannii*. The discovery of an intracellular niche for this pathogen, as well as hyperinvasive isolates, may help guide the development of antimicrobial therapies and diagnostics in the future.

**KEYWORDS** *Acinetobacter baumannii*, clinical isolates, high-content screen, intracellular multiplication

*A*cinetobacter baumannii is a nosocomial pathogen posing a growing global health threat due to its remarkable ability to persist in the environment and acquire extensive multidrug resistance. In some countries, carbapenem resistance rates have surpassed 80% (1), ranking this pathogen as a top priority for developing new antibiotics by the World Health Organization (2). Carbapenem resistance is associated mostly with eight international clonal (IC) lineages (3). Although community-acquired cases have been described, *A. baumannii* mainly impacts patients with severe underlying disease such as those in intensive care units. One of the most frequent clinical manifestations of *A. baumannii* infection is ventilator-associated pneumonia (VAP), often associated with a poor prognosis. Of increasing concern is the recent appearance of hypervirulent strains that present concurrently extensive antibiotic resistance and have been implicated in hospital and animal infection outbreaks, of which some were fatal (4–6).

Despite its growing importance, the mechanisms underlying *A. baumannii* virulence remain poorly characterized. Its ability to adhere to abiotic surfaces and form biofilms enables colonization of medical equipment and surfaces (7). Adherence to human cells and the interplay with innate immune cells have also proven critical to *A. baumannii* virulence (8, 9).

*A. baumannii* is primarily considered an extracellular pathogen. In some studies, clinical isolates were described as noninvasive in human lung epithelial cell lines (10). *A. baumannii* laboratory and clinical strains were also shown to be rapidly phagocytosed and killed by cultured macrophages and neutrophils (11, 12). However, previous studies have highlighted the ability of different *A. baumannii* strains to be internalized or to actively invade host cells (13–20). Intracellular survival of *A. baumannii* in cultured cells has been reported when critical antibacterial host response pathways were inhibited, such as Nod1/Nod2, nitric oxide, or autophagy (12, 14, 15). A few recent studies have suggested that some strains of *A. baumannii* can invade and transiently survive within epithelial human cells and macrophages (16–18, 20, 21). Although the *A. baumannii* strain ATCC 19606 is killed by macrophages, it was shown to enter epithelial cells by a zipper-like mechanism associated with actin microfilaments and microtubules (16). Similarly, the *A. baumannii* strain ATCC 17978 can survive within human epithelial lung cells, resulting in activation of lysosomal biogenesis and autophagy (18). More recently, the strain *A. baumannii* AB5075-UW was also shown to invade nonphagocytic cells by binding carcinoembryonic antigen-related cell adhesion (CECAM) molecules (17). Once intracellular, AB5075-UW survives within a vacuole associated with early and late endosomal GTPases Rab5 and Rab7 as well as the autophagy protein light chain 3 (LC3). Nevertheless, bacteria are progressively killed by vacuolar acidification (17).

In this work, we highlight several *A. baumannii* clinical isolates that multiply intracellularly within large late endosomal-derived vacuoles without autophagy features. We found that this intracellular multiplication is not associated with cytotoxicity. High-content screening of 43 clinical isolates suggests that a significant proportion of isolates are capable of invasion and intracellular survival, with a minor subset able to establish intracellular replication niches. Notably, a few isolates were hyperinvasive and hyperreplicative. Taken together, these results shed some light on a potentially clinically relevant intracellular niche for some *A. baumannii* isolates, which could impact patient management in a hospital setting, provide a target for new therapeutic approaches, and constitute a biomarker for virulent strains.

## RESULTS

**The hypervirulent *A. baumannii* C4 clinical strain is able to invade human A549 lung epithelial cells.** Given the increasing reports on hypervirulent strains of *A. baumannii* in hospitals, we set out to investigate if this enhanced virulence could be attributed to particular interactions with host cells. We initially focused on the *A. baumannii* strain C4 belonging to the international clonal lineage 4 (IC4), isolated from a wound swab from a patient hospitalized in Germany in 2010. This strain was transmitted between 4 patients who subsequently died, and we hypothesized that this strain was hypervirulent. We first assessed the virulence of this strain *in vitro* using the well-established *Galleria mellonella* infection model. The experimental infection revealed that C4 is significantly more virulent than the well-characterized *A. baumannii* strain ATCC 17978, used as a control in this study (Fig. 1A). To investigate the mechanisms underlying the virulence of the *A. baumannii* C4 strain, we infected human lung epithelial A549 cells and quantified the levels of adhesion at 1 h postinfection (hpi). Analysis of the percentage of bacterial adhesion relative to the inocula following enumeration of viable bacteria was equivalent, for both revealed that the C4 strain does not have enhanced adhesion capacity (Fig. 1B). We next measured the levels of lactate dehydrogenase (LDH) released from cells infected for 6 h with *A. baumannii* C4 in comparison with ATCC 17978, the environmental *A. baumannii* strain DSM 30011, and the cytotoxic *Pseudomonas aeruginosa* strain PA14. Cells were either infected for 1 h and then incubated with antibiotics to eliminate extracellular bacteria or washed to remove nonadherent bacteria without the use of antibiotics. No significant cytotoxicity was observed in cells infected with any of the *A. baumannii* strains tested, in contrast to *P. aeruginosa* (Fig. 1C).

We next assessed the production of the proinflammatory interleukin 6 (IL-6) by A549 cells infected with *A. baumannii* C4, ATCC 17978, and DSM 30011 at 3, 8, and 24 hpi. All strains induced an increase in IL-6 secretion over time. *A. baumannii* ATCC 17978 induced IL-6 secretion with a peak at 24 hpi (Fig. 1D). Similar levels of IL-6 were detected in cells infected with the environmental *A. baumannii* isolate DSM 30011. The C4 strain also induced an equivalent increase in IL-6 secretion at 3 and 8 h, although it induced lower levels at 24 hpi (Fig. 1D).

Finally, we tested whether *A. baumannii* C4 can invade human nonphagocytic cells. We observed a clear condensation of actin around the bacteria upon entry into the cell of the *A. baumannii* C4 strain, suggesting that actin cytoskeleton rearrangements are involved in the invasion process (Fig. 1E) as previously described for *A. baumannii* ATCC 19606 (16). At 24 hpi, confocal microscopy revealed intracellular clusters of bacteria suggestive of intracellular replication (Fig. 1F). Z-stack analysis of infected cells labeled with tubulin confirmed the intracellular nature of these bacterial clusters (Fig. 1F). In summary, the *A. baumannii* C4 strain can invade host cells and form intracellular bacterial clusters without causing cell lysis.

**Clinical *A. baumannii* strains C4 and ABC141 multiply in human cells.** To determine if the presence of intracellular bacterial clusters at 24 hpi resulted from intracellular replication by the *A. baumannii* C4 strain, we quantified, by microscopy, the number of bacteria per cell at 1 and 24 hpi for the *A. baumannii* C4 and ATCC 17978 strains. We initially tested two bacterial growth conditions, exponential and stationary. Equivalent results were obtained for both bacterial growth phases, so only data referring to exponential growth are shown, and we selected this condition for all subsequent experiments. At 1 h, we observed that infected cells contained one or two bacteria per cell for either the *A. baumannii* C4 and ATCC 17978 strains (Fig. 2A) with equivalent percentages of infected cells observed (Fig. 2B) and equivalent rates of uptake quantified by differential labeling of intracellular and extracellular bacteria by microscopy (Fig. 2C). At 24 hpi, we could detect the appearance of bacterial clusters with multiple bacteria (up to 20 bacteria per cluster) for the *A. baumannii* C4 strain in contrast to the *A. baumannii* ATCC 17978, which remained with only a few bacteria per cell (Fig. 2A and D). Taken together, these results indicate that the virulent *A. baumannii* C4 strain is capable of intracellular multiplication.

We next expanded our study to another clinical isolate available in the laboratory, the *A. baumannii* ABC141 strain, that represents IC5 and was isolated from a skin swab.

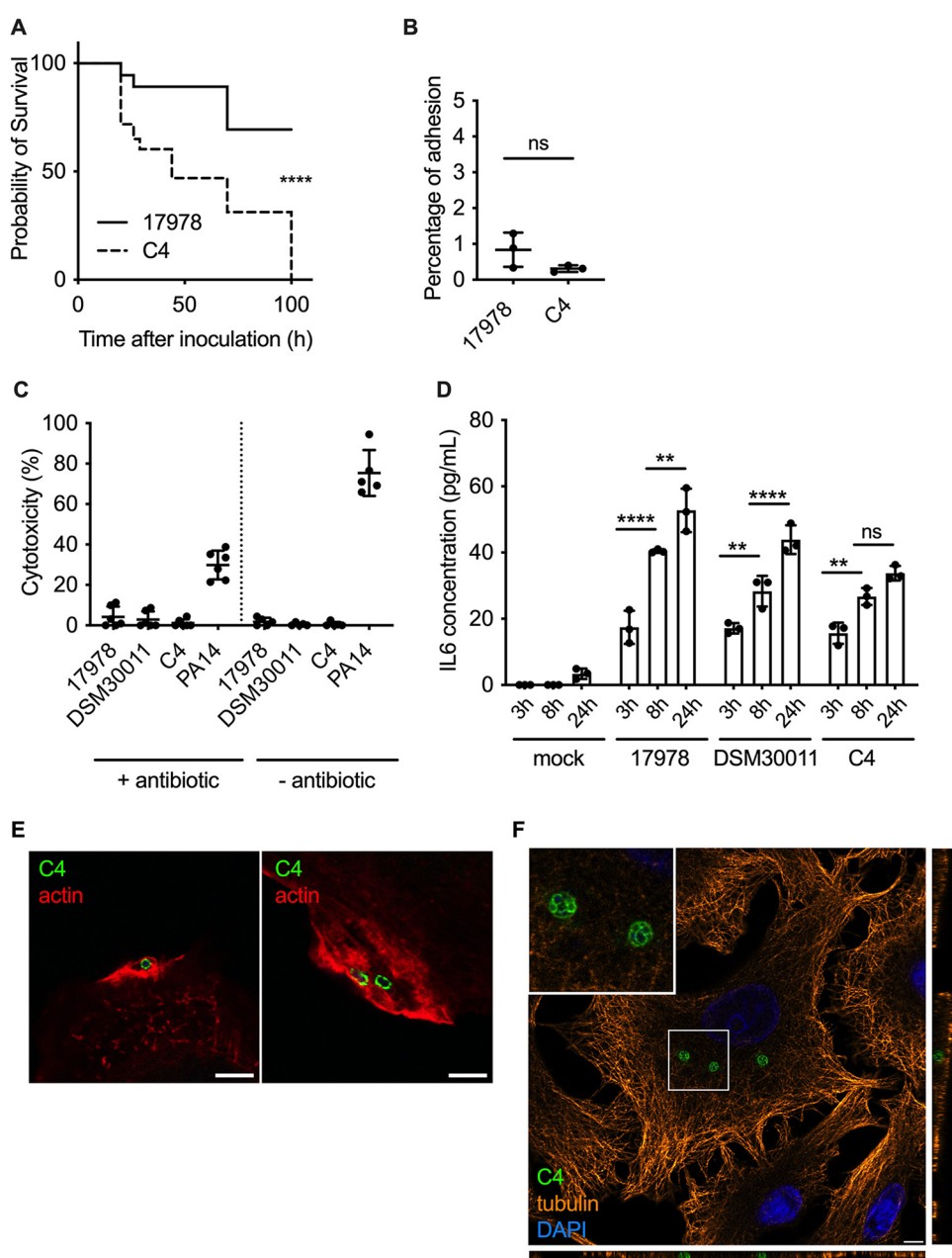

**FIG 1** Characterization of the hypervirulent *A. baumannii* C4 clinical strain. (A) Kaplan-Meier survival curves were generated from *G. mellonella* injected with the *A. baumannii* strains C4 or ATCC 17978 ($1 \times 10^6$ CFU per insect). Mortalities were counted regularly over 100 h. *A. baumannii* C4 is significantly more virulent than ATCC 17978. ****, $P < 0.0001$ (log-rank test). Data are representative of 3 independent experiments. (B) Percentage of *A. baumannii* C4 and ATCC 17978 adhesion to A549 cells (MOI of 100) obtained after enumeration of viable bacteria before and after 1 h of infection. Data correspond to mean $\pm$ SD and are from 3 independent experiments. Their percentages of adhesion are not significantly different. (C) Cytotoxicity of *A. baumannii* ATCC 17978, DSM 30011, C4, and *P. aeruginosa* PA14 were monitored using LDH assay in A549 cells in the presence or absence of antibiotic. No cytotoxicity was observed for any *A. baumannii* strains. Data correspond to mean $\pm$ SD from 5 independent experiments. (D) Quantification of IL-6 concentration produced by A549 cells infected by *A. baumannii* ATCC 17978, DSM 30011, and C4 at 3 h, 8 h, and 24 hpi. Statistical comparison was done with two-way ANOVA with a Holm-Sidak's correction for multiple comparisons. ****, $P < 0.0001$ between *A. baumannii* ATCC 17978 3 h and 8 h; **, $P < 0.01$ between DSM 30011 3 h and 8 h; **, $P < 0.01$ between C4 3 h and 8 h; **, $P < 0.01$ between ATCC 17978 8 h and 24 h; ****, $P < 0.0001$ between DSM 30011 8 h and 24 h. Not all comparisons are shown. Data correspond to mean $\pm$ SD from 3 independent experiments. (E) A549 cells were infected with *A. baumannii* C4, immunolabeled 1 hpi, and analyzed using confocal immunofluorescence microscopy. *A. baumannii* (green) was labeled with specific antibodies, and phalloidin was used to visualize actin (red). Scale bars correspond to 5 $\mu$m. (F) A549 cells were infected with *A. baumannii* C4 and labeled for tubulin (orange), *A. baumannii* (green), and the nucleus with DAPI (blue). The orthogonal view of the z-stack shows that *A. baumannii* C4 is able to enter into the cell and form intracellular bacterial clusters.

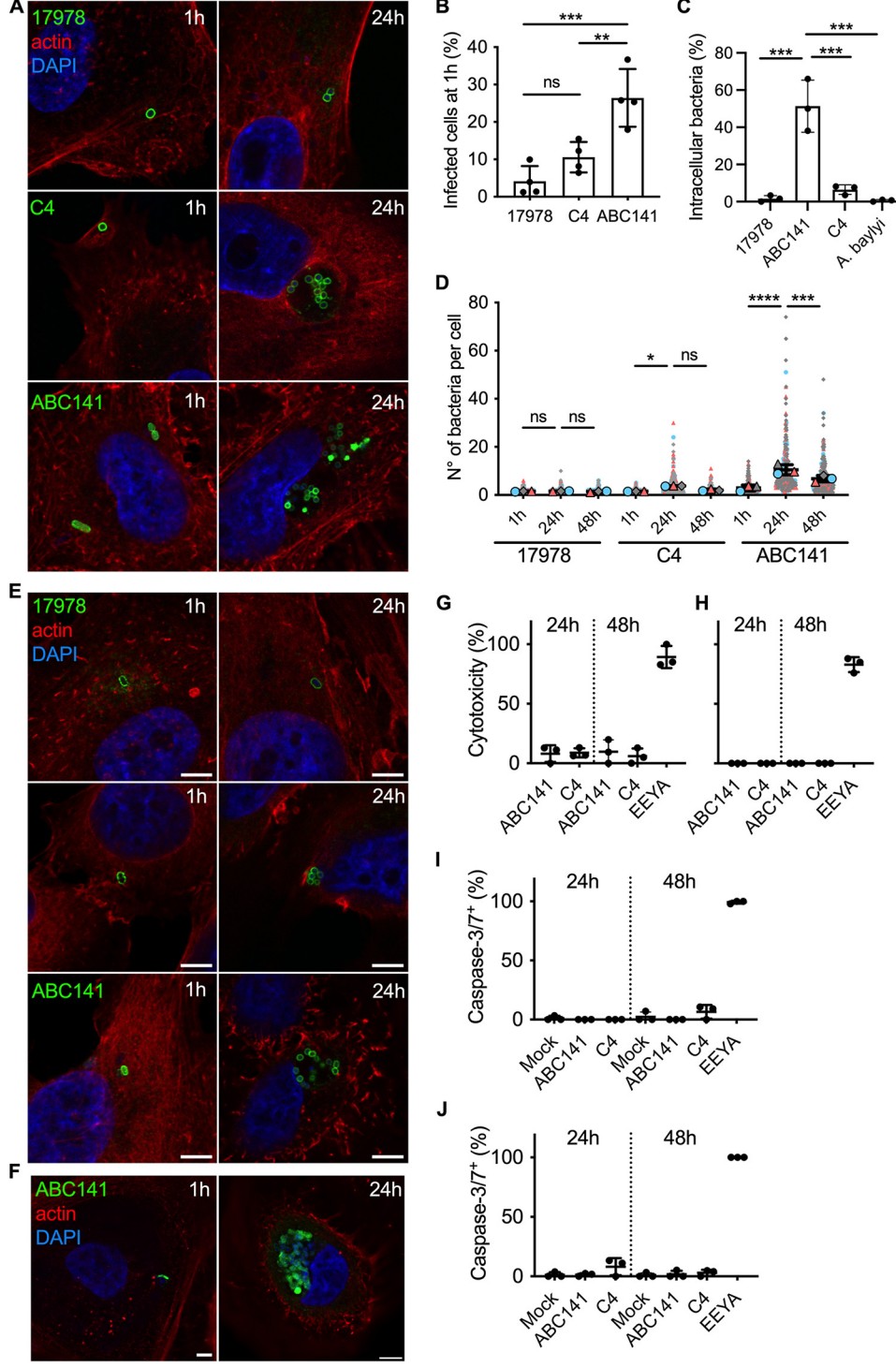

**FIG 2** *A. baumannii* C4 and ABC141 strains multiply intracellularly. Human cells were infected with *A. baumannii* strains ATCC 17978, C4, or ABC141 and analyzed using confocal immunofluorescence microscopy. (A) Infected A549 cells were fixed at 1 and 24 hpi and labeled with phalloidin and DAPI to visualize actin cytoskeleton (red) and nucleus (blue), respectively. *A. baumannii* strains were labeled with specific antibodies (green). Representative images are shown. Scale bars correspond to 5 μm. (B) Percentage of A549 cells infected by *A. baumannii* strains ATCC 17978, C4, or ABC141 quantified by microscopy. Data correspond to mean ± SD and are from 4 independent experiments, counted by confocal microscopy. One-way ANOVA with a Holm-Sidak's correction was used for multiple comparisons. **, $P < 0.01$ between C4 and ABC141; ***, $P < 0.001$ between *A. baumannii* strains ATCC 17978 and ABC141; ns, $P > 0.05$ between ATCC 17978 and C4. (C) Quantification of the percentage of intracellular bacteria at 1 h postinfection by differential labeling of intracellular and extracellular bacteria. At least 100 bacteria were counted per experiment and condition, and

A much higher percentage of cells were infected with *A. baumannii* ABC141 at 1 hpi than *A. baumannii* C4 and ATCC 17978 (Fig. 2B), suggesting ABC141 is hyperinvasive. Consistently, a much higher rate of uptake was observed for *A. baumannii* ABC141 in relation to the other strains (Fig. 2C). Importantly, at 24 hpi, we observed a significant increase in the number of bacteria per cell (Fig. 2A and D), indicative of extensive intracellular replication. Interestingly, we often visualized multiple bacterial clusters per cell (Fig. 2A), with some clusters containing up to 50 bacteria. At 48 hpi, however, a decrease in the numbers of *A. baumannii* ABC141 clusters was observed (Fig. 2D). It is important to note that when A549 cells were infected with a stationary-phase culture of ABC141, we did not observe hyperinvasion nor significant intracellular replication, suggesting that the growth stage is critical to confer hyperinvasive and replicative phenotypes to this strain, in contrast to the *A. baumannii* C4 strain.

To determine whether *A. baumannii* strains C4 and ABC141 are able to multiply within other cell types, we infected human endothelial EA.hy 926 cells. Similar to A549 cells, all three strains invaded EA.hy 926 cells, with a few bacteria per cell visible at 1 hpi, but only *A. baumannii* C4 and ABC141 were able to multiply intracellularly (Fig. 2E). We next infected primary human keratinocytes with the most invasive *A. baumannii* strain ABC141 to test its ability to multiply intracellularly in primary cells rather than immortalized cells lines. Extensive *A. baumannii* ABC141 multiplication was observed, confirming this phenotype is not cell type specific (Fig. 2F).

Because of the high numbers of intracellular bacteria observed for *A. baumannii* ABC141 at 24 hpi, we next investigated whether heavily infected cells displayed signs of cell death. We first monitored total LDH release, and less than 20% toxicity was observed in A549 (Fig. 2G), whereas none was detected in EA.hy 926 cells (Fig. 2H) up to 48 h. As previous reports suggest that *A. baumannii* can induce apoptosis (22–24), we monitored caspase 3 and 7 activation in infected cells by microscopy at 24 and 48 hpi in comparison to mock-infected cells and cells treated with a high concentration of eeyarestatin as a positive control. Neither *A. baumannii* C4 nor ABC141 induced significant caspase-dependent cell death in A549 or EA.hy 926 cells (Fig. 2I and J).

**Intracellular replicative strains of *A. baumannii* multiply within large nonacidic vacuoles positive for LAMP1.** We next set out to characterize the nature of these *A. baumannii* intracellular compartments. We first labeled infected cells with the nonspecific lectin wheat germ agglutinin-fluorescein isothiocyanate conjugate (WGA-FITC) to visualize cellular membranes. This fluorescent probe labels sialic acid and glycoproteins containing $\beta$-(1→4)-*N*-acetyl-D-glucosamine, such as cellulose, chitin, and peptidoglycans. The majority of the intracellular bacterial clusters of *A. baumannii* C4 and ABC141 were surrounded by a WGA-positive membrane in A549 epithelial cells (Fig. 3A), suggesting enrichment in host cell sialic acid and/or bacterial glycoproteins on the vacuolar membrane surrounding replicating bacteria. To further characterize these *Acinetobacter*-containing vacuoles (ACVs), we labeled A549 infected cells for the late

**FIG 2 Legend (Continued)**
data correspond to mean ± SD from 3 independent experiments. (D) The numbers of intracellular bacteria per cell were counted at 1, 24, and 48 hpi and represented in "superplot." Colors (gray, blue, and pink) correspond to 3 independent experiments. Each cell counted is shown (small dots) together with the means of each experiment (larger dots), which were used for statistical analysis. Statistical comparison was done with two-way ANOVA with a Holm-Sidak's correction for multiple comparisons. ns, $P > 0.05$ between *A. baumannii* strains ATCC 17978 1 h and 24 h and ATCC 17978 24 h and 48 h for both comparisons; *, $P < 0.05$ between C4 1 h and 24 h; ns; $P > 0.05$ between C4 24 h and 48 h; ****, $P < 0.0001$ between ABC141 1 h and 24 h; ***, $P < 0.001$ between ABC141 24 h and 48 h. (E) Infected EA.hy 926 cells were fixed at 1 and 24 hpi and labeled with phalloidin and DAPI to visualize actin cytoskeleton (red) and nucleus (blue), respectively. *A. baumannii* strains were labeled with specific antibodies (green). Representative images are shown. Scale bars correspond to 5 $\mu$m. (F) Human primary keratinocytes were infected with ABC141 for 24 h and immunolabeled with phalloidin (red), DAPI (blue), and an anti-*Acinetobacter* antibody (green). Scale bars correspond to 5 $\mu$m. (G and H) Cytotoxicity was monitored using LDH assay in A549 (G) and EA.hy 926 cells (H) following infection with the indicated strains. (I and J) Quantification of the percentage of infected A549 (I) and EA.hy 926 (J) cells that are caspase3/7 positive following infection with *A. baumannii* C4 or ABC141 for 24 and 48 h. Noninfected cells were included as a negative control, and cells treated and incubated with eeyarestatin for 5 h were used as positive control. Data correspond to mean ± SD from 3 independent experiments.

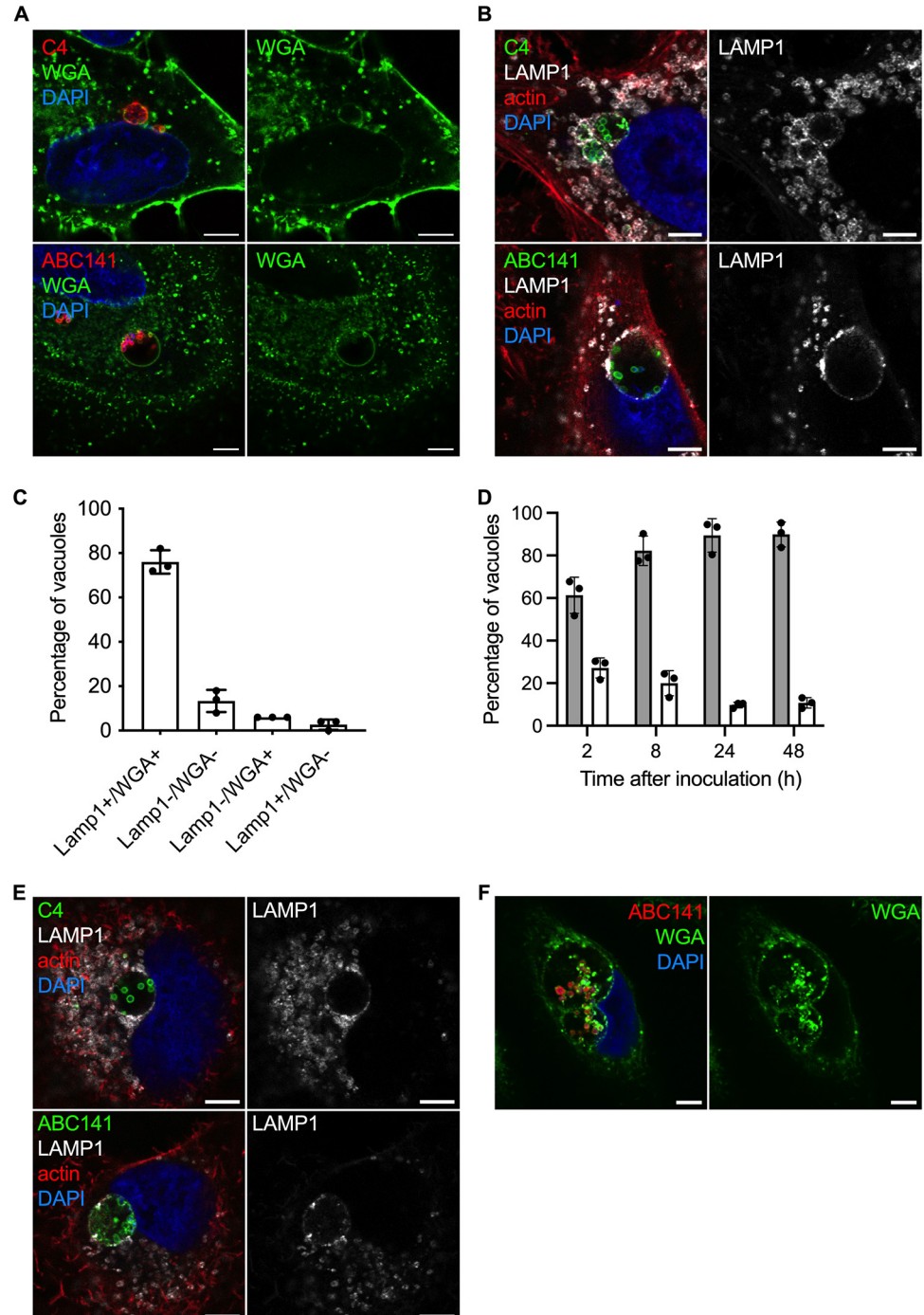

**FIG 3** Multiplication of *A. baumannii* C4 and ABC141 occurs in large vacuoles positive for LAMP1. Human cells were infected with *A. baumannii* C4 or ABC141, immunolabeled 24 hpi, and analyzed using confocal immunofluorescence microscopy. Representative images are shown. Scale bars correspond to 5 $\mu$m. (A) Membranes of A549 cells were labeled with WGA (green), the nucleus with DAPI (blue), and *A. baumannii* with antibodies (red). *A. baumannii* C4 and ABC141 multiply in *Acinetobacter*-containing vacuoles (ACVs). (B) A549 cells were labeled with anti-LAMP1 antibodies (gray) with phalloidin and DAPI to visualize actin cytoskeleton and nucleus, respectively, and *A. baumannii* isolates were labeled with specific antibodies (green). ACVs are positive for LAMP1 in A549 cells. (C) The numbers of ACVs positive for LAMP-1 and/or WGA staining were counted at 24 hpi for cells infected by *A. baumannii* ABC141. ACVs positive for both LAMP1 and WGA staining represent 78% of intracellular bacteria clusters. Data correspond to mean ± SD from 3 independent experiments. (D) A549 cells infected by *A. baumannii* ABC141 were labeled with Lysotracker DND-99 (white bars) or LAMP1 (gray bars) at 2, 8, 24, and 48 h postinfection to visualize acidic compartments and the kinetics of acquisition of the late endosomal marker LAMP1. The majority of ACVs do not show features of acidic lysosomes. Data correspond to mean ± SD from 3 independent experiments. (E) EA.hy 926 cells were labeled with anti-LAMP1 antibody (gray), anti-*A. baumannii* specific antibodies (green), and with phalloidin and DAPI to visualize actin cytoskeleton and nucleus. ACV are positive for LAMP1 in EA.hy 926 cells. (F) Representative images of human primary keratinocytes infected with *A. baumannii* ABC141 for 24 h. Nuclei were labeled with DAPI (blue), ABC141 were labeled with a specific antibody (red), and membranes with WGA (green).

endosomal marker lysosome-associated membrane glycoprotein 1 (LAMP1). For both *A. baumannii* C4 and ABC141, ACVs were decorated with LAMP1, suggesting these are late endosomal or lysosomal-derived vacuoles (Fig. 3B). The proportion of LAMP-positive ACVs was quantified using the most invasive strain of *A. baumannii* ABC141. Over 75% of ACVs were positive for WGA and LAMP1, confirming the vast majority of intracellular multiplying bacteria are within membrane-bound compartments (Fig. 3C). LAMP1 was detected on ACVs from 2 to 48 h of infection, but only a small percentage of ACVs were positive for lysotracker at the late stages of the infection (Fig. 3D), suggesting that this strain is inhibiting acidification or lysosomal fusion. Of note, 13% of intracellular bacterial clusters were not labeled (Fig. 3C), which could correspond to cytosolic bacteria.

To assess if *A. baumannii* C4 and ABC141 ACVs were WGA and LAMP1 positive in other cell types, we infected EA.hy 926 cells and primary keratinocytes for 24 h (Fig. 3E and F). Microscopic analysis confirmed that both strains multiply in equivalent intracellular compartments in endothelial cells and primary keratinocytes. Indeed, *A. baumannii* C4 and ABC141 multiply in very large vacuoles whose size could exceed 7 $\mu$m, in some cases seemingly "pressing" against the nucleus (Fig. 3E and F).

Taken together, these results indicate that the intracellular replicative strains of *A. baumannii* C4 and ABC141 multiply in LAMP1-positive nonacidic vacuoles in human nonphagocytic cells.

***Acinetobacter*-containing vacuoles have a single membrane and do not colocalize with the autophagy marker LC3.** In view of the large size of ACVs and the presence of LAMP1, we next hypothesized that these could correspond to autophagosomes, as previously demonstrated surrounding intracellular *A. baumannii* strain AB5075 before bacterial killing (17). To test this hypothesis, we immunolabeled A549-infected cells for the microtubule-associated protein 1A/1B-light chain 3 (LC3), widely used as a marker of autophagy. We did not observe any LC3 labeling associated with *A. baumannii* C4 or ABC141 ACVs (Fig. 4A). To confirm that ACVs were not autophagy-derived vacuoles, we performed transmission electron microscopy on A549 cells infected with ABC141, which shows a higher rate of invasion. We observed *A. baumannii* ABC141 exclusively within large vacuoles composed of a single membrane, confirming that ACVs are not autophagosomes. Moreover, we noted that ACVs contain free space and multiple small vesicles (Fig. 4B).

In summary, we found that intracellular replicative strains of *A. baumannii* can create a niche favorable to their multiplication within human cells.

**Prevalence of intracellular replicative strains in clinical *A. baumannii* isolates.** Since with this small selection of *A. baumannii* clinical isolates, we observed several distinct phenotypes, ranging from noninvasive to hyperinvasive and replicative, we gathered a larger collection of 43 nonduplicate clinical isolates to determine the prevalence of each phenotype. IC4 and IC5 isolates were chosen to compare with *A. baumannii* C4 and ABC141, respectively, while isolates representing other clonal lineages and sporadic isolates were chosen to compare across the lineages (Fig. 5; Table S1 in the supplemental material). To screen a high number of isolates, we set up a high-content screen to image infected A549 cells at 24 hpi labeled for LAMP1 to clearly distinguish intracellular bacterial clusters. This was combined with a mix of three antibodies against *A. baumannii*, which we confirmed beforehand could label all bacteria tested. We classified the observed phenotypes in 4 categories, (i) noninvasive, (ii) capable of entry and survival, (iii) capable of replication and visible by the formation of LAMP1-positive bacterial clusters, and (iv) hyperinvasive, with a rate of infection equivalent to the *A. baumannii* ABC141 strain. In total, 46 isolates were screened in two independent assays, including *A. baumannii* ATCC 17978, C4, and ABC141. From this collection, 24 isolates were noninvasive, with no bacteria detected intracellularly, whereas 18 isolates were capable of cell entry but did not show any intracellular multiplication, giving a phenotype equivalent to the *A. baumannii* ATCC 17978. The five *A. baumannii* isolates, C4, ABC141, BMBF_193, ABC020, and R10, were capable of intracellular multiplication in LAMP1-positive vacuoles, but only the *A. baumannii* ABC141 and BMB_193 isolates were also hyperinvasive. These results highlight the variety of phenotypes observed

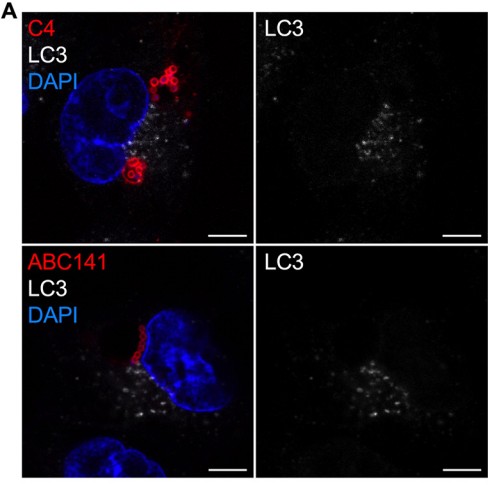

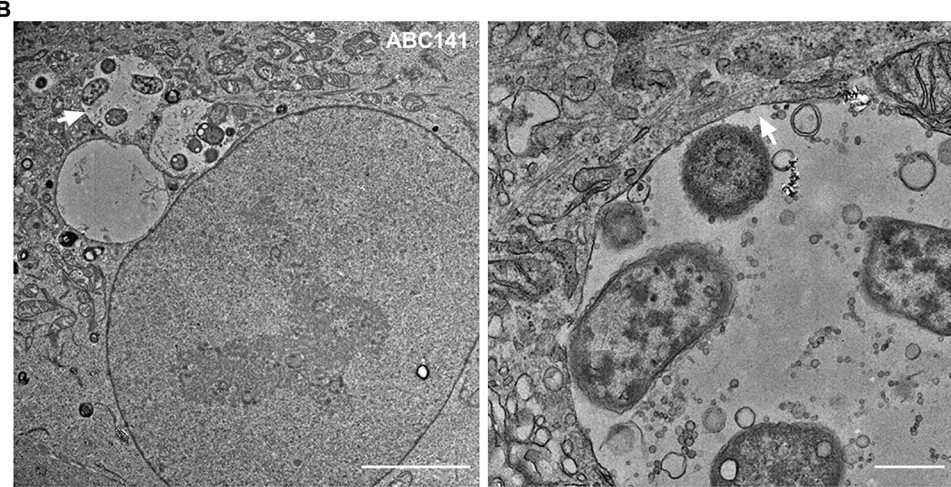

**FIG 4** ACVs are single-membrane vacuoles that do not colocalize with the autophagy marker LC3. (A) A549 cells were infected with *A. baumannii* C4 or ABC141, immunolabeled 24 h postinfection, and analyzed using confocal immunofluorescence microscopy. *A. baumannii* (red) and LC3 (white) were labeled with antibodies and nucleus with DAPI (blue). ACVs do not colocalize with LC3. Representative images are shown. Scale bars correspond to 5 μm. (B) Transmission electron microscopy of infected A549 cells by *A. baumannii* ABC141 24 h postinfection. The scale bar corresponds to 5 μm. The second picture represents a zoom of the vacuole indicated by the white arrow. The scale bar corresponds to 500 nm. *A. baumannii* ABC141 multiplies in ACVs with a single membrane.

for clinical *A. baumannii* isolates and confirm that a subset of these has the ability to invade and multiply within host cells.

## DISCUSSION

In this study, we highlight a diversity of phenotypes exhibited by *A. baumannii* clinical isolates regarding their interaction with human nonphagocytic cells. The observed interactions were strain specific, and there was no correlation within clonal lineages. A subset of these isolates, including one that was associated with a high fatality rate, are capable of active intracellular replication without induction of cytotoxicity. This highlights an important niche that could be hindering treatment of patients infected with these types of isolates.

Although considered originally as an extracellular pathogen, a growing number of studies have shown that some strains of *A. baumannii* are able to invade epithelial and endothelial human cells (13–20, 23, 25–28). *A. baumannii* cell invasion occurs via a zipper-like mechanism involving actin microfilaments and microtubules (13, 19) and is dependent on clathrin, *β*-arrestins, and phospholipase C-coupled G-proteins, as their

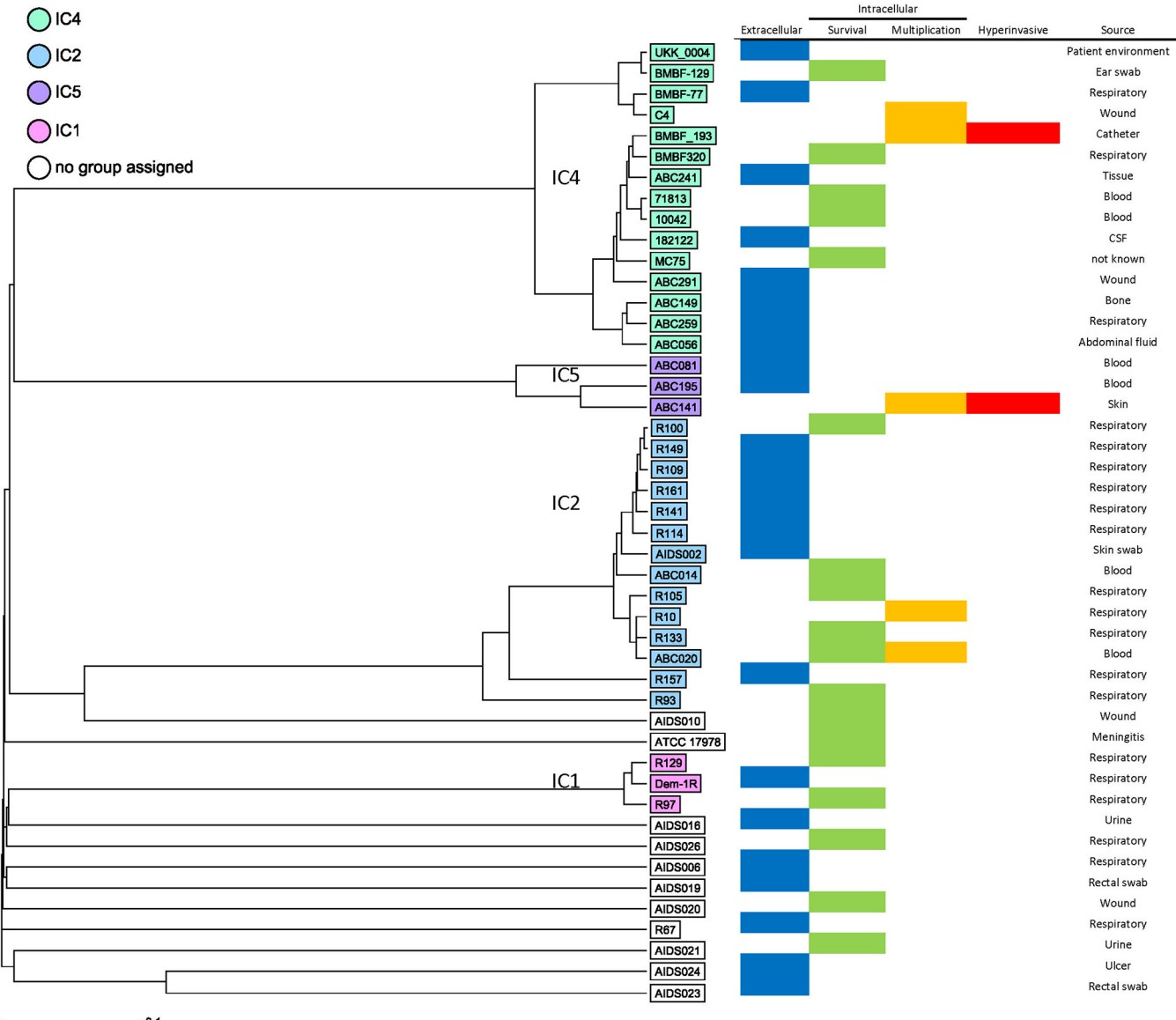

**FIG 5** Unweighted pair group method using average linkages (UPGMA) tree generated using core genome multilocus sequence typing (cgMLST) data showing the results of a high-content screen by confocal microscopy of clinical *A. baumannii* isolates. A549 cells were infected with 43 clinical *A. baumannii* isolates and *A. baumannii* C4, ABC141, and ATCC 17978 strains as controls. Cells were fixed at 24 hpi and immunolabeled with a mix of three anti-*Acinetobacter* antibodies, LAMP1, and DAPI. For each strain, z-stack series were imaged at five different positions of the well. cgMLST was performed using Ridom SeqSphere+ using 2,390 targets. Isolates are colored based on clustering with a clonal lineage. Those within white boxes do not cluster with any of the lineages. The source of the isolates is indicated.

inhibition prevents its entry into the cell (26). The platelet-activating factor receptor (PAFR) was also shown to enhance bacterial internalization following interaction with the phosphorylcholine-containing outer membrane protein of *A. baumannii* (26). However, this interaction results in an increased intracellular calcium concentration and subsequent cell death. The use of random mutant libraries highlighted the importance of *A. baumannii* phospholipases in cellular invasion (25). The outer membrane proteins OmpA and Omp33 of *A. baumannii* have also been implicated, but the cellular interactions taking place remain unknown (13, 19). A diversity of phenotypes regarding the ability to invade cells has been previously reported (16). The discovery of hyperinvasive strains of *A. baumannii* described in our study will provide an excellent tool to decipher the underlying mechanisms. All of these bacterial factors implicated in internalization are encoded within the genome of *A. baumannii* ABC141, suggesting

enhanced invasion could be due to different regulatory mechanisms or, alternatively, to the presence of new bacterial factors involved that have yet to be identified. Detailed genome and transcriptome comparisons are necessary to decipher the virulence factors involved.

Although all these data strongly support the existence of an *A. baumannii* intracellular phase, the fate of intracellular bacteria remains less well characterized. The intracellular persistence of *A. baumannii* in cultured cells was described in both epithelial cells and macrophages (11, 13–18). In most cases, bacteria were found to transiently survive within an acidic and autophagy-derived compartment, which eventually results in efficient bacterial killing (16). In our study, we found several *A. baumannii* clinical isolates capable of extensive intracellular multiplication for up to 24 and 48 hpi. Importantly, these observations were done by microscopy, which allowed us to ensure that the increase in bacterial numbers is not due to extracellular replication of bacteria that are not efficiently killed by the antibiotics. We have observed, for a few antibiotic-sensitive and -resistant strains excluded from this study, the formation of bacterial aggregates tightly adhered to the surface of the cells and the coverslips that seem protected from antibiotic treatment, which would mislead CFU counts. Interestingly, other studies have identified clinical strains able to multiply in cultured macrophages by bacterial CFU counts (21), suggesting these phenotypes could be extended to phagocytic cells. Consistently, while this work was being submitted, a study has shown that *A. baumannii* urinary tract infection isolates were capable of intracellular replication in cultured macrophages (29) by both CFU and microscopy counts, strengthening the hypothesis that an intracellular replication niche may be clinically relevant for a subset of *A. baumannii* isolates.

In this study, we found that *A. baumannii* C4 and ABC141 multiply within membrane-bound vacuoles, reaching 20 to 70 bacteria at 24 hpi, not previously observed in *A. baumannii*–infected cells. ACVs are very large compartments surrounded by a glycoprotein-rich membrane which colocalizes with LAMP1. Further work has to be done to determine if the glycoproteins decorating ACVs are from eukaryotic or bacterial origin or both. The presence of LAMP1, indicative of a late endosomal-derived compartment, was not accompanied by acidification for the majority of ACVs, as only a few were lysotracker positive. These results are in contrast with previous reports showing persistence of some *A. baumannii* within LAMP1 acidic compartments, notably for the *A. baumannii* strain ATCC 19606 (20). It has been shown that *A. baumannii* multiplies slower in acidic pH environment *in vitro* and the inhibition of acidification by bafilomycin A1 increases the number of intracellular *A. baumannii* AB5075 organisms (17, 18). Therefore, we hypothesize that our *A. baumannii* clinical isolates are able to establish a niche suitable for intracellular replication by preventing fusion with degradative lysosomes and blocking their acidification. Interestingly, these vacuoles attain very large sizes, even pushing the nucleus of the cell. To our knowledge, this type of vacuole is not commonly observed for intracellular pathogens and merits further investigation. One example of a pathogen reported to create large multivesicular vacuoles with the appearance of "empty space" reminiscent of these ACVs is *Helicobacter pylori* (30). This pathogen colonizes human stomach mucosa, causes gastrointestinal diseases, and multiplies in human cells. It has been shown that intracellular *H. pylori* can modulate autophagy, blocking the acidification of the vacuoles and multiplying inside large autophagosomes (31).

Autophagy is a key cellular process for cell survival and host innate immunity that participates in the elimination of invading bacteria. Not surprisingly, many pathogens have developed mechanisms to modulate autophagy or hijack this process in order to promote their survival and multiplication. In the case of *A. baumannii* infection, there are a few reports implicating autophagy during infection. Indeed, the *A. baumannii* ATCC 17978 strain was reported to persist within double-membrane vacuoles (23). The porin Omp33-36 was the virulence factor implicated in induction of apoptosis via caspase activation and induction of autophagy visible by the accumulation of p62 and

LC3B-II (23). *A. baumannii* infection itself was shown to induce Beclin-1-dependent autophagy via the AMPK/ERK signaling pathway (20) with the involvement of the porin OmpA (15). In addition, the transcriptional factor EB (18) was shown to block acidification of the autophagosome-lysosome system and enhance bacterial persistence. In contrast to all these reports, ACVs described in this study did not show any LC3 immunolabeling. Furthermore, transmission electron microscopy confirmed that ACVs are single-membrane vacuoles and therefore not autophagic in nature. We conclude that the *A. baumannii* clinical isolates capable of intracellular multiplication presented in this study create specialized replicative vacuoles that successfully escape autophagy and subsequent lysosomal degradation. After 24 h of replication, in the case of the *A. baumannii* ABC141 strain, we observed a reduction in the presence of large vacuoles. The recent work of the Feldman lab showed that *A. baumannii* urinary tract infection strains capable of intramacrophage multiplication (29) escape from the cell. Therefore, it is likely that a similar event is taking place for *A. baumannii* C4 and ABC141 in epithelial cells, allowing bacteria to multiply without killing the host cell and then egress from infected cells to disseminate within the tissue or systemically.

The intracellular replication phenotype we observed seems to be common to several clinical isolates, suggesting this phenotype is relevant from a clinical point of view. In addition to the 3 initial strains studied, 43 isolates were tested using a high-content microscopy screen, allowing us to get some insight into the prevalence of this phenotype. We found that the ability to invade and persist without multiplication was quite frequent among the different isolates tested, but this was irrespective of their clonal lineages, suggesting strain-specific traits that have yet to be identified. Importantly, 5 strains in total in our study were capable of intracellular replication, all within large LAMP1-positive vacuoles, suggesting that a significant proportion of current *A. baumannii* isolates share this intracellular niche. It is important to keep in mind that we may be underestimating the prevalence of this phenotype due to the specificity of the conditions tested. For example, some *A. baumannii* isolates may have tissue specificity or require particular growth conditions. Consistently, an overnight culture of *A. baumannii* ABC141 significantly reduces its invasion capacity, while this is not the case for *A. baumannii* C4.

Interestingly, intracellular multiplication was observed for both the virulent *A. baumannii* C4 strain and also for *A. baumannii* ABC141, which was isolated from the skin. This result suggests that the ability to establish an intracellular niche is not the direct cause of enhanced virulence in patients. This is not surprising, as the severity of an *A. baumannii* infection is tightly connected to the host susceptibility, and hence, it is not possible to assign levels of virulence to different *A. baumannii* strains based on the site of isolation. It is possible that virulence properties are randomly distributed among clinical isolates, and their presence or absence does not necessarily translate into clinical pathogenicity. Nonetheless, an intracellular phase could confer enhanced protection to the pathogen against some antibiotics, promote dissemination in the host, or give rise to relapsing infections. This phenomenon has been described in another nosocomial pathogen, *Staphylococcus aureus*, which is also able to survive and multiply inside host cells (32, 33). The invasion of human THP-1 macrophages protects *S. aureus* from vancomycin, oxacillin, moxifloxacin, rifampicin, gentamicin, and oritavancin (34). In a mouse model, intracellular *S. aureus* can establish an infection even in the presence of vancomycin (35). Furthermore, *S. aureus* internalized in keratinocytes are not killed by antibiotics even at 20-fold their MIC (36). Similar observations were made with invasive *P. aeruginosa* strains that are protected from aminoglycosides, in contrast to $\beta$-lactams, fluoroquinolones, and colistin (37). Therefore, the intracellular nature of specific *A. baumannii* isolates may have important clinical consequences. The protection conferred by the intracellular environment could aggravate antibiotic resistance of *A. baumannii* in patients, which will not be detected by antibiotic susceptibility testing *in vitro*. Discriminating intracellularly replicating isolates may provide an important diagnostic tool in the future. Our work is a first step in the identification of

hyperinvasive and replicative isolates, providing insight on their intracellular trafficking, which may ultimately prove beneficial to help adapt antimicrobial therapies against this nosocomial pathogen.

## MATERIALS AND METHODS

**Culture of cell lines and primary cells.** A549 (human epithelial lung cell line) and EA.hy 926 (human endothelial somatic cell line) cells were bought from Merck and ATCC, respectively. Both cell lines were grown in Dulbecco's modified Eagle medium (DMEM) supplemented with 1% L-glutamine and 10% fetal calf serum at 37°C with 5% $CO_2$ atmosphere.

Normal human keratinocyte (NHK) cultures were established from foreskin after dermal-epidermal dissociation with 0.05% trypsin and 0.01% ethylenediaminetetraacetic acid (EDTA) in phosphate-buffered saline (PBS), pH 7.4, and grown in supplemented keratinocyte growth medium containing 0.15 mM CaCl2 (KBM-2 BulletKit; Lonza Biosciences, Basel, Switzerland) as previously described (38). NHK was used between passages 1 to 3. Biopsy specimens were obtained following ethical and safety guidelines according to French regulation donors (declaration no. DC-2008-162 delivered to the Cell and Tissue Bank of Hospices Civils de Lyon).

**Bacterial strains and culture conditions. (i) Bacterial strains, culture conditions, and whole-genome sequencing.** The *A. baumannii* ATCC 17978 strain was obtained from ATCC and all other clinical isolates of *A. baumannii* from the University of Cologne (Table S1) in the supplemental material. *Acinetobacter* isolates were grown on lysogenic broth (LB) agar, pH 7.4, for 24 h at 37°C. For liquid cultures, a single colony was inoculated in LB agar, pH 7.4, and incubated for 17 h for stationary-phase cultures for the experiments described in Fig. 1. All other experiments were done with late exponential cultures in which a colony was inoculated in LB overnight (16 h) and then diluted 1:100 and grown until an optical density at 600 nm ($OD_{600}$) of 0.8 to 1 was reached.

Whole-genome sequencing and analysis were performed as previously described (39). The raw sequencing reads used in this project were submitted to the European Nucleotide Archive (https://www.ebi.ac.uk/ena/), and their nucleotide accession numbers are listed in Table S1.

**Human cells infection and adhesion assays.** Human cells were grown in 96-well culture plates at $1.6 \times 10^4$ cells/well, 24-well culture plates at $4 \times 10^4$ cells/well, 12-well culture plates at $2.5 \times 10^5$ cells/well, and 6-well culture plates at $1 \times 10^6$ cells/well for the high-throughput screen, fluorescence microscopy, adhesion assays, and electron microscopy, respectively. Then, they were infected at a multiplicity of infection (MOI) of 100 *A. baumannii* isolates in supplemented DMEM medium prewarmed at 37°C. Plates were centrifuged at $400 \times g$ for 5 min and incubated for 1 h at 37°C with 5% $CO_2$ atmosphere. Once incubated, plates were washed 5 times with PBS to remove extracellular bacteria.

**(i) Adhesion assays.** Cells were lysed by incubation for 5 min with 0.1% sodium deoxycholate. Bacteria were enumerated before and after infection to calculate the percentage of *A. baumannii* adhesion. The MOI was verified by the number of CFU/mL in the inocula.

**(ii) Fluorescence microscopy.** Infected cells were incubated in supplemented DMEM medium with apramycin (40 $\mu$g/mL) or tobramycin (50 $\mu$g/mL) for 24 h or 48 h. After 24 h of incubation, the antibiotic was changed followed by a second incubation of 24 h at 37°C with 5% $CO_2$ atmosphere (time point, 48 h). Coverslips were fixed at each time point (1 h, 24 h, and 48 h) with Antigenfix (Diapath; paraformaldehyde, pH 7.2 to 7.4) for 15 min or methanol (precooled at $-20$°C) for 5 min at room temperature. Finally, samples were washed with PBS 5 times and kept at 4°C.

**(iii) Electron microscopy.** Samples were fixed in 2% glutaraldehyde and 2% paraformaldehyde in 0.1 M cacodylate buffer (pH 7.2) overnight at 4°C. After extensive washing in 0.1 M cacodylate at 4°C, cells were postfixed with 2% osmium tetroxide in 0.1 M cacodylate buffer for 1 h at 4°C. After rinsing with water, a contrast was performed with 1% uranyl acetate. The samples were then dehydrated using a graded series of ethanol and embedded in Epon resin. After polymerization at 60°C for 48 h, ultrathin sections (60 nm) were cut using a Leica UC7 microtome and contrasted with lead citrate. Samples were examined with a Jeol 1400 Flash transmission electron microscope.

**Immunolabeling.** Once fixed, cells were incubated in PBS with 1% saponin and 2% bovine serum albumin (BSA) for 1 h at room temperature for permeabilization and blocking. Primary antibodies were diluted in the blocking solution and incubated for 2 h. To label bacteria, we raised antibodies in rabbits against *A. baumannii* ATCC 17978, AB5075, and C4, using a lyophilized preparation and a speedy immunization polyclonal program (Biotem, France; Eurogentec, Belgium). The anti-*A. baumannii* antibody mix was diluted at 1:1,000 for clinical isolates and at 1:10,000 for the *A. baumannii* ATCC 17978 strain. The following additional antibodies were used: mouse anti-LC3 (1:1,000), mouse anti-$\beta$ tubulin (1:200), and mouse anti-LAMP1 H4A3 (1:200) from the Developmental Studies Hybridoma Bank, created by the NICHD of the NIH and maintained at the University of Iowa. Coverslips were then washed twice in PBS, 0.1% saponin, and 2% BSA. Secondary antibodies and dyes, anti-rabbit Alexa Fluor 488 or 555 (1:500), anti-mouse Alexa Fluor 555 or 647 (1:500), wheat germ agglutinin (WGA)-FITC conjugate (Sigma; 1:200), phalloidin-Atto 647N (1:200), and DAPI (4',6-diamidino-2-phenylindole) nuclear dye (Bio-Rad;1:1000) were diluted in the blocking solution. Cells were incubated for 1 h followed by two washes in PBS with 0.1% saponin/2% BSA, one wash in PBS, and one wash in distilled water. Finally, coverslips were mounted with ProLong Gold (Thermo Fisher).

For the high-content screening, cells were labeled with LAMP1 and a mix of the 3 homemade antibodies used in this study, which enabled labeling of all the isolates. DAPI was also included.

For differential microscopy analysis of intracellular and extracellular bacteria, staining was first done without permeabilization (primary and secondary antibodies) followed with a second stage of staining

Rubio et al.

in the presence of a permeabilizing agent as described above, using a secondary antibody with a different fluorochrome.

**Caspase 3/7 detection.** A549 cells were infected for 24 h as described above. Noninfected cells incubated with eeyarestatine (500 $\mu$M) for 24 h were used as positive control. Each condition was next incubated with the Caspase 3/7 Green detection reagent (CellEvent kit) diluted in PBS with 5% fetal calf serum (FCS) for 30 min. Cells were fixed with Antigenfix for 15 min at room temperature. Finally, samples were washed with PBS 5 times and kept at 4°C.

**Lysotracker.** A549 cells were infected for 22 h as described above. They were then incubated for 2 h with the Lysotracker DND-99 (75 nM) in DMEM medium prewarmed at 37°C. Finally, the loading solution was replaced by Antigenfix to fix cells. Labeled cells were immediately observed by confocal microscopy.

**Counting the number of infected cells and intracellular bacteria.** The percentage of infected cells at 1 or 24 h postinfection was calculated by counting the number of noninfected and infected cells per 10 fields for *A. baumannii* ATCC 17978, C4, ABC141, and ABC56 for 3 independent experiments. To ensure intracellular bacteria were quantified, phalloidin labeling was done to mark the actin cytoskeleton, and z-stack analysis was performed.

The number of intracellular bacteria per cell for *A. baumannii* ATCC 17978, C4, and ABC141 strains were counted for 3 independent experiments. Results are represented in "superplot" as described by Lord and colleagues (40), in which each experiment is color coded. All the cells counted are presented as the corresponding mean and standard deviation. Statistical analysis was done by comparing the means of independent experiments.

The number of vacuoles with a LAMP1 and/or WGA labeling were counted for *A. baumannii* ABC141 in 3 independent experiments.

**Confocal microscopy.** For all images and counting, coverslips were mounted with ProLong Gold (Thermo Fisher) and observed with a Zeiss LSM800 Airy Scan laser scanning confocal microscope with a 63× oil immersion objective. For the high-throughput screen, images were collected with a Yokogawa CQ1, and a 40× objective was used. Finally, they were analyzed with Fiji (41) and assembled in Figure J (42).

**Experimental infection (*Galleria*).** *G. mellonella* larvae, purchased from Sud Est Appats (http://www.sudestappats.fr/), were used within 48 h of arrival. *A. baumannii* isolates tested were grown overnight in LB and then diluted with PBS to enable injection of $1 \times 10^6$ CFU, as determined by CFU plating. Bacterial suspensions were injected into the hemolymph of each larva (second last left proleg) using a Hamilton syringe (10 $\mu$L). Groups of 20 randomly picked larvae were used for each isolate. Survival curves were plotted using GraphPad, and comparisons in survival were calculated using the log-rank Mantel-Cox test.

**IL-6 quantifications.** A549 cells were grown in 96-well culture plates at a density of $1 \times 10^5$ cells/mL. The next day, the cells were infected with the different *A. baumannii* isolates from a culture in stationary phase diluted to an MOI of 100. The infection protocol is the same as described above. The supernatants were recovered after 8, 2,4, or 48 h of infection. The concentration of IL-6 was quantified by enzyme-linked immunosorbent assay (ELISA) (Human IL-6 ELISA Ready-Set-Go!; Thermo Fisher) by following the supplier's protocol.

**Statistical tests.** All data sets were tested for normality using Shapiro-Wilkinson test. When a normal distribution was confirmed we used a one-way analysis of variance (ANOVA) test with a Holm-Sidak's correction for multiple comparisons. For two independent variables, a two-way ANOVA test was used. For data sets that did not show normality, a Kruskal-Wallis test was applied, with Dunn's correction. All analyses were done using Prism GraphPad 7.

## SUPPLEMENTAL MATERIAL

Supplemental material is available online only.

**FIG S1**, XLSX file, 0.01 MB.

**TABLE S1**, XLSX file, 0.01 MB.

## ACKNOWLEDGMENTS

We acknowledge the contribution of the SFR Biosciences (UAR3444/CNRS, US8/INSERM, ENS de Lyon, UCBL) imaging facility Plateau Technique Imagerie/Microcopie (PLATIM) and the Centre Technologique des Microstructures, Université Lyon 1. We are grateful to Matthias Faure (CIRI, Lyon) for providing us with the anti-LC3 antibody and advice. We thank Carina Müller (DZIF, Cologne) for her technical assistance with the ABC clinical isolates.

This work was funded by the Fondation pour la Recherche Médicale grant DEQ20180339215. S.P.S. is an INSERM researcher. P.G.H. and S. Göttig were supported by the Deutsche Forschungsgemeinschaft (DFG FOR 2251).

T.R. carried out and analyzed all experiments with *A. baumannii* ABC141 and performed quantifications of intracellular multiplication and trafficking of all isolates. S. Gagné characterized the *A. baumannii* C4 strain and discovered the intracellular multiplication phenotype. C.D. carried out and analyzed the caspase experiments and high-content screening. D.M. set up and managed the clinical collection and optimized the experimental protocol for the screening. C.Dias assisted T.R. with infection

experiments. C.C. did the electron microscopy, and P.R. isolated and cultured the keratinocytes. H.S., P.G.H., and S. Göttig provided all the clinical isolates and input regarding the epidemiology, genomics, and interpretation of the screen results. The project was conceived, supervised, and funded by S.P.S. The manuscript was written by T.R. and S.P.S. All authors read and approved the manuscript.

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
