## [Reviewer comments · mSystems]

Incidence of an intracellular multiplication niche amongst *Acinetobacter baumannii* clinical isolates

Tristan Rubio, Stéphanie Gagné, Charline Debruyne, Chloé Dias, Caroline Cluzel, Doriane Mongellaz, Patricia Rousselle, Stephan Göttig, Harald Seifert, Paul Higgins, and Suzana Salcedo

Corresponding Author(s): Suzana Salcedo, CNRS, Université de Lyon

Review Timeline:

Submission Date:	April 20, 2021
Editorial Decision:	July 22, 2021
Revision Received:	November 4, 2021
Editorial Decision:	November 29, 2021
Revision Received:	December 15, 2021
Accepted:	January 5, 2022

Editor: Kim Barrett

Reviewer(s): The reviewers have opted to remain anonymous.

Transaction Report:

DOI: <https://doi.org/10.1128/mSystems.00488-21>

July 22, 2021

Dr. Suzana Pinto Salcedo
CNRS, Université de Lyon
MMSB, UMR5086
7 Passage du Vercors
Lyon 69367
France

Re: mSystems00488-21 (Incidence of an intracellular multiplication niche amongst *Acinetobacter baumannii* clinical isolates)

Dear Dr. Suzana Pinto Salcedo:

Thank you for submitting your manuscript to mSystems. We have completed our review and I am pleased to inform you that, in principle, we expect to accept it for publication in mSystems. However, acceptance will not be final until you have adequately addressed the reviewer comments.

The editor truly regrets the significant delays that have occurred during the review of this manuscript

Preparing Revision Guidelines

For complete guidelines on revision requirements for your article type, please see the journal Article Types requirement at <https://journals.asm.org/journal/mSystems/article-types>. **Submissions of a paper that does not conform to mSystems guidelines will delay acceptance of your manuscript.**

Sincerely,

Kim Barrett

Editor, mSystems

Journals Department
Reviewer comments:

Reviewer #1 (Comments for the Author):

Rubia and coworkers report an exciting finding: the invasion and intracellular survival of *A. baumannii* in non phagocytic cells. An important strength of the study is the test of a significant number of clinical strains to demonstrate how widespread the phenotype it is. This work will catalyze significant research by other groups, and may have clinical implications as some of treatments used in the clinic do not target intracellular bacteria.

Major comments.

1. Fig 1D. The limited scope of the results (only one cytokine and not assessment of the activation of signalling pathways) makes necessary for the authors to tone down the comparison between strains. This reviewer urges authors to just indicate that the strains do elicit IL6, and that the levels increase over time. And that this result is not related to the number of intracellular bacteria. In fact, this is the interesting observation.
2. An interesting observation across the manuscript is the apparent localization of the ACV near the nucleus. Could the authors quantify this?
3. A549 cells are notoriously robust cells. This reviewer will urge authors to test toxicity in other cell types using LDH released (or similar test).
4. Although it is used in this manner in papers, LAMP1 cannot be regarded as a lysosomal marker. These comments need to be removed from the text, and just refer to a late endosomal marker.
5. It would be appropriate to test the colocalization of the ACV with a lysosomal marker over time.
6. Lamp1, and lysotracker colocalization with the ACV should be assessed over time.
7. It would have been interesting to get a sense of the cellular elements implicated on *A. baumannii* engulfment (actin cytoskeleton, microtubules, PI3K,...)

Reviewer #2 (Comments for the Author):

In the manuscript by Rubio, et al., the authors investigate whether a subset of *Acinetobacter baumannii* isolates are able to enter and grow within normally nonphagocytic cells. They provide evidence that an isolate thought to be of increased virulence, C4, enters and grows within cells. Entry and growth occur without attendant death, and was also observed with the isolate ABC141, which appeared to be hyperinvasive. The compartment surrounding bacterial compartments was LAMP1-positive but appeared to be nonacidic. Similarly, there was no evidence for autophagy, in contrast with previous reports regarding *A. baumannii*. In a screen of over 40 isolates coming from various lineages showed intracellular multiplication of 4 of them and two that were hyperinvasive.

The manuscript is an interesting contribution to the *Acinetobacter* literature. My main issues with the manuscript are that nonstandard assays are used to measure and compare adhesion to uptake, and there is no real negative control to compare the data to. For instance, I believe that in Fig. 1B, the authors are challenging cells at an MOI =100, and they are seeing about 0.2% cell associated bacteria based on viable counts (this should be made clear). If they did the experiment with just *E. coli* K12 that is nonpiliated (a standard nonadhesive control) is this significantly different, given those high MOI conditions? In addition, standard uptake assays involve either differential antibody staining (before and after permeabilization of fixed cells) or aminoglycoside survival after uptake. The latter they may have performed, since they mentioned that in fluorescent microscopy they incubated cells with apramycin or tobramycin, but it is not clear that all the bacteria were sensitive to these antibiotics. The authors should also discuss whether there is any evidence in animal models for bacteria residing in nonphagocytic cells. It's possible this might occur in the urinary tract.

Detailed comments:

1. Fig. 1A: The authors state that C4 is significantly more virulent than AB17978, but it is impossible to evaluate this statement given the nature of the experiment. Is there some statistical test to show this is actually significant?
2. Fig. 1B: Please state more clearly how this experiment is done and that the quantitation is performed differently from Fig.2B. It took considerable time to figure out whether the difference was due to assay measures or cell line differences.
3. Fig. 1b, 2B. You really need *E. coli* K12 as a negative control here.
4. Comparing Fig. 1B to 2B: Why is it that there more C4 associated in Fig. 2B than 17978, but the number is reversed in Fig. 1B?
5. Fig. 2B and throughout: A standard measure of uptake should be used, as described in the general comments (antibody protection or aminoglycoside protection).
6. Figure 2B, hyperinvasion of ABC141 and other strains: from the *Staphylococcus* literature, hyper-cell association can be attributed to loss of capsule. Particularly with ABC141, and the hyperinvasive strains in Fig. 5, is this due to absence of capsule production? For instance, with ABC141, it's possible that capsule is only produced in post-exponential phase, since uptake only occurs during exponential growth.

7. Fig. 5: AB17978 is listed as a urine isolate. Just want to make sure this is correct.
8. Very minor point: I am curious why the authors chose clonal lineage analysis rather than MLST
9. Fig. 5 is very useful, but there are issues with some of these assays and the antibiotics used to kill extracellular bacteria. Are all of these bacteria sensitive to the antibiotics used (tobramycin or apramycin)?

Dear Editors,

Please find below our point-by-point answer to the reviewers.

We have also noticed a mistake in the Figure 5 for one of the strains (ABC020) from the screen that is capable of intracellular multiplication. This is now corrected. In addition, small edits were made throughout the text to correct for typos and misuse of the word “strain” vs “isolate”.

Reviewer #1

1. Fig 1D. The limited scope of the results (only one cytokine and not assessment of the activation of signalling pathways) makes necessary for the authors to tone down the comparison between strains. This reviewer urges authors to just indicate that the strains do elicit IL6, and that the levels increase over time. And that this result is not related to the number of intracellular bacteria. In fact, this is the interesting observation.

The text has been modified accordingly.

2. An interesting observation across the manuscript is the apparent localization of the ACV near the nucleus. Could the authors quantify this?

This is indeed the case for a significant proportion of infected cells presenting large clusters but seems to only be occurring at 24h. We are currently investigating if this location corresponds to the MTOC and if vacuole localization is dependent on microtubules. However, as this is still very preliminary and requires live-imaging for confirmation we prefer not to include this aspect in the manuscript.

3. A549 cells are notoriously robust cells. This reviewer will urge authors to test toxicity in other cell types using LDH released (or similar test).

We have expanded our cytotoxicity analysis as recommended. We have now analysed both A549 and EA.hy 926 endothelial cells by measuring LDH release but also caspase activation to enable single cells analysis (of infected cells only), at both 24 and 48h post-infection. The results are now included in Figures 2G-J and the text modified.

4. Although it is used in this manner in papers, LAMP1 cannot be regarded as a lysosomal marker. These comments need to be removed from the text, and just refer to a late endosomal marker.

The text was modified accordingly.

5. It would be appropriate to test the colocalization of the ACV with a lysosomal marker over time.

These data are now included in Figure 3D and the legends and text modified.

6. Lamp1, and lysotracker colocalization with the ACV should be assessed over time.

These data can be found in Figure 3D and the legends and text modified.

7. It would have been interesting to get a sense of the cellular elements implicated on A. baumannii engulfment (actin cytoskeleton, microtubules, PI3K,...)

In this manuscript we did not focus on entry as there are several reports implicating actin and microtubules in entry of different *A. baumannii* strains into non-phagocytic cells (for example, Choi et al 2008). We are currently constructing fluorescent strains to extend these studies for the hyperinvasive strain ABC141.

Reviewer #2:

*My main issues with the manuscript are that nonstandard assays are used to measure and compare adhesion to uptake, and there is no real negative control to compare the data to. For instance, I believe that in Fig. 1B, the authors are challenging cells at an MOI =100, and they are seeing about 0.2% cell associated bacteria based on viable counts (this should be made clear). If they did the experiment with just *E. coli* K12 that is nonpiliated (a standard nonadhesive control) is this significantly different, given those high MOI conditions?*

We understand the confusion and have now clarified the text. The experiment presented in Figure 1B is a standard adhesion assay, measuring number of CFUs left after extensive washing at 1h post-infection. We have now modified the text and figure legend. There are many publications showing *A. baumannii* adheres to host cells, although, in our hands this remains at low levels. An *E. coli* K12 shows no adhesion (Figure A for reviewers only). Increased MOI induces higher total CFU counts (Figure A for the reviewers only) but it does not impact the percentage of adhesion.

Figure A. Adhesion assay quantifying viable CFU counts at 1h post-infection of A549 cells. Total CFU counts are presented (instead of normalized to the inocula) to allow comparison of the different MOIs. Strains 17978, C4 and non-adhesive *E. coli* K12 strain was used as control.

In addition, standard uptake assays involve either differential antibody staining (before and after permeabilization of fixed cells) or aminoglycoside survival after uptake. The latter they may have performed, since they mentioned that in fluorescent microscopy they incubated cells with apramycin or tobramycin, but it is not clear that all the bacteria were sensitive to these antibiotics.

To quantify the percentage of cells with intracellular bacteria we used microscopy analysis of cells labeled with phalloidin, allowing to establish intracellular location in relation to the actin cytoskeleton. This is now clarified in the text. We have carried out the suggested experiment, with differential labeling and the results are now included in Figure 2C. We do not feel CFU counts for intracellular bacteria are adapted as they do not allow for single cell analysis, which is now the gold-standard in the field. For example, they do not allow distinguishing a few heavily infected cells from many cells with only a few bacteria.

The authors should also discuss whether there is any evidence in animal models for bacteria residing in nonphagocytic cells. Its possible this might occur in the urinary tract.

As probably this reviewer is aware from his/her suggestion, beautiful unpublished work from

Hultgren's lab suggests this is indeed the case, a reservoir of replicating intracellular bacteria can be found in epithelial cells of the urinary tract in a mouse model of infection. We believe that it is not for us to carry out these experiments and compete with our colleagues and we will await the publication of their results.

Detailed comments:

1. Fig. 1A: *The authors state that C4 is significantly more virulent than AB17978, but it is impossible to evaluate this statement given the nature of the experiment. Is there some statistical test to show this is actually significant?*

The statistical analysis (log-rank test) was originally referred to in the Figure legend. We have now added the information to the Figure P= ****.

2. Fig. 1B: *Please state more clearly how this experiment is done and that the quantitation is performed differently from Fig.2B. It took considerable time to figure out whether the difference was due to assay measures or cell line differences.*

We have now modified the text to explain that Fig 1B corresponds to CFU counts and Fig 2B microscopy counts.

3. Fig. 1b, 2B. *You really need E. coli K12 as a negative control here.*

The adhesion of *A. baumannii* is well established in the field. *E. coli* K12 showed no adhesion (Figure A of this letter) and was therefore not an ideal control for invasion. Instead we have included *A. baylyi*, which shows equivalent adhesion to 17978 but for which we do not see any invasion (Fig. 2C).

4. Comparing Fig. 1B to 2B: *Why is it that there more C4 associated in Fig. 2B than 17978, but the number is reversed in Fig. 1B?*

In Figure 1B the difference between 17978 and C4 is not statistically significant. This is also the case for Figure 2B, as indicated in the figures.

5. Fig. 2B and throughout: *A standard measure of uptake should be used, as described in the general comments (antibody protection or aminoglycoside protection).*

Aminoglycoside protection assays have been frequently used in the past and might be useful in some circumstances. However, besides technical issues, they are less informative than single cell analysis techniques. E.g. one can have a single cell with 200 bacteria or 200 cells with 1 bacterium each and obtain the same CFU counts; yet the result is very different. Single cell analysis by microscopy or image-coupled cytometry are more precise methods. We have chosen a microscopy-based approach in this study.

6. Figure 2B, hyperinvasion of ABC141 and other strains: *from the Staphylococcus literature, hyper-cell association can be attributed to loss of capsule. Particularly with ABC141, and the hyperinvasive strains in Fig. 5, is this due to absence of capsule production? For instance, with ABC141, its possible that capsule is only produced in post-exponential phase, since uptake only occurs during exponential growth.*

This is an excellent comment and something we are starting to investigate. As we are not experts in capsule biology we are establishing the appropriate collaborations to do so. This is however, in our opinion, beyond the scope of this manuscript.

7. Fig. 5: *AB17978 is listed as a urine isolate. Just want to make sure this is correct.*

Apologies for the mistake. This has now been corrected.

8. *Very minor point: I am curious why the authors chose clonal lineage analysis rather than MLST*

We have performed several large global epidemiological studies with *Acinetobacter baumannii*, and have found that there are nine distinct lineages that are widely distributed throughout the world, i.e. they are international. Sequence types are good to cluster isolates, but there are two competing schemes, and so two different nomenclatures. Because of the international makeup of our isolates, we prefer to group them using the international clones nomenclature.

9. *Fig. 5 is very useful, but there are issues with some of these assays and the antibiotics used to kill extracellular bacteria. Are all of these bacteria sensitive to the antibiotics used (tobramycin or apramycin)?*

The antibiotic data sensitivity is now included as Supplementary Fig1.

November 29, 2021

Dr. Suzana Pinto Salcedo
CNRS, Université de Lyon
MMSB, UMR5086
7 Passage du Vercors
Lyon 69367
France

Re: mSystems00488-21R1 (Incidence of an intracellular multiplication niche amongst *Acinetobacter baumannii* clinical isolates)

Dear Dr. Suzana Pinto Salcedo:

Thank you for submitting your manuscript to mSystems. We have completed our review and I am pleased to inform you that, in principle, we expect to accept it for publication in mSystems. However, acceptance will not be final until you have adequately addressed the reviewer comments.

The reviewers and editor believe that the manuscript is much improved. However, the authors are asked to address the outstanding concern of reviewer 2 that it is unexpected that no adhesion of the control strain would be observed.

Preparing Revision Guidelines

Sincerely,

Kim Barrett

Editor, mSystems

Journals Department
Reviewer comments:

Reviewer #1 (Comments for the Author):

The revised version of the manuscript by Rubio and colleagues is significantly improved. Authors have addressed the most outstanding concerns in a rigorous manner.
This reviewer does not have additional concerns.

Reviewer #2 (Comments for the Author):

The authors have answered my queries sufficiently. I will point out that having done many adhesion studies with various nonpiliated E. coli strains, absolutely no binding of this control is totally unexpected. Even in the absence of cells, there is expected to be some low level binding of E. coli to plastic.

Dear Editors, and Reviewers,

Please find below our point-by-point answer to the reviewers.

Reviewer#2

1. The authors have answered my queries sufficiently. I will point out that having done many adhesion studies with various nonpiliated *E. coli* strains, absolutely no binding of this control is totally unexpected. Even in the absence of cells, there is expected to be some low level binding of *E. coli* to plastic.

We have traced the precise strain used in the experiment presented in our previous letter to the reviewers and found that it corresponds to a non-fimbriated and non-hemagglutinated HB101 strain. Probably, with the extensive washes we are doing in our assay and the specific characteristics of this strain we lost all detectable adhesion, explaining our results.

We hence performed new adhesion experiments employing another K12 derivative *E. coli* J53 (Yi et al. J Bacteriol. 2012; 194) with A549 cells (MOI 100, 2 h infection). We compared the *A. baumannii* strains ATCC 17978 and C4 strains (both used in our paper) with *A. baumannii* ATCC 19606, *E. coli* K12 J53, and a 19606 mutant strain lacking the *ata* gene, which is essential for adhesion (Weidensdorfer et al., Virulence 2019). Adhesion of *E. coli* K12 J53 and *A. baumannii* ATCC 19606 showed almost identical adhesion rates albeit at lower levels than *A. baumannii* wild-type 17978 and C4 strains, which as reported in our manuscript have comparable adhesion levels (Figure 1b). As expected, the mutant lacking *Ata* is the least adhesive and in fact this constitutes the best control for our experiments.

We were not aware that we were initially working with a very particular derivative of *E. coli* K12 and will no longer use it in our adhesion assays. We thank the reviewer for bringing this to our attention.

January 5, 2022

Dr. Suzana Pinto Salcedo
CNRS, Université de Lyon
MMSB, UMR5086
7 Passage du Vercors
Lyon 69367
France

Re: mSystems00488-21R2 (Incidence of an intracellular multiplication niche amongst *Acinetobacter baumannii* clinical isolates)

Dear Dr. Suzana Pinto Salcedo:

Thanks for responding appropriately to the remaining issue that was outstanding. The editor apologizes sincerely for the delay in rendering this final decision.

Your manuscript has been accepted, and I am forwarding it to the ASM Journals Department for publication. For your reference, ASM Journals' address is given below. Before it can be scheduled for publication, your manuscript will be checked by the mSystems senior production editor, Ellie Ghatineh, to make sure that all elements meet the technical requirements for publication. She will contact you if anything needs to be revised before copyediting and production can begin. Otherwise, you will be notified when your proofs are ready to be viewed.

Publication Fees:

We recognize that the video files can become quite large, and so to avoid quality loss ASM suggests sending the video file via <https://www.wetransfer.com/>. When you have a final version of the video and the still ready to share, please send it to mssystemsjournal@msubmit.net.

Sincerely,

Kim Barrett
Editor, mSystems

Journals Department
Supplemental Material: Accept
Supplemental Material: Accept